# Plant-pollinator interactions along the altitudinal gradient in *Berberis lycium* royle: An endangered medicinal plant of the Himalayan region

Nahila Anjum[1], Sajid Khan[1], Susheel Verma[2], Kailash S. Gaira[3], Balwant Rawat[4], Nakul Chettri[5]*, Mohd Hanief[1]*

1 Department of Botany, School of Biosciences and Biotechnology, Baba Ghulam Shah Badshah University, Rajouri, Jammu and Kashmir, India, 2 Department of Botany, University of Jammu, Jammu and Kashmir, India, 3 G. B. Pant National Institute of Himalayan Environment, Sikkim Regional Centre, Pangthang, Gangtok, Sikkim, India, 4 School of Agriculture, Graphic Era Hill University, Dehradun, Uttarakhand, India, 5 International Centre for Integrated Mountain Development, Kathmandu, Nepal

* nchettri2001@gmail.com & haniefdu@gmail.com

## Abstract

The terrestrial ecosystem, particularly mountain regions, influences species distribution by providing diverse climatic conditions that vary with rising altitude. These climatic factors play a significant role in determining species phenology and niche width. However, the environmental factors influencing pollination dynamics of specific plant species across altitudes remain unexplored. Considering the gaps, we assess how the composition and abundance of pollinator fauna associated with the important medicinal plant *Berberis lycium* Royle (Berberidaceae) vary across five distinct altitudinal gradients (800–2200 m) in the Pir-Panjal mountain range in the northwestern part of the Indian Himalayan region. Pollinators, including bees, butterflies, wasps, and flies were monitored over two consecutive flowering seasons (2022–2023). A total of 39 insect species representing five orders and 17 families, were recorded visiting *B. lycium* during its flowering period across the altitudinal range. The linear regression model indicated that all four pollination indices exhibited a declining trend with increasing altitude when data were pooled together. However, only foraging speed (FS) and index of visiting rate (IVR) were showed significant declines. Among individual pollinator groups, only Lepidoptera displayed a significant relationship with altitude, while other groups exhibited asynchrony along the altitudinal gradient. Furthermore, reproductive output (fruit and seed production) declined significantly with increasing altitude. Our findings suggest that while altitude influences species distribution but also differentially shapes plant-pollinator interactions, pollinator foraging behaviour, and reproductive success. This study highlights the importance of monitoring plant-pollinator interactions in fragile Himalayan ecosystem, where environmental changes could have cascading effects on ecological stability.

**Data availability statement:** All relevant data are within the paper and its Supporting Information files.

**Funding:** The author(s) received no specific funding for this work.

**Competing interests:** The authors have declared that no competing interests exist

## 1. Introduction

The terrestrial ecosystem, particularly mountain region, is extremely fragile influenced by numerous factors that affect species diversity and distribution patterns. This region is home to nearly half of the world's biodiversity hotspots and supports numerous endemic species [1].The Himalaya possesses unique bio-climatic geographical features in Asia, forming a natural barrier between the Tibetan plateau and the lowland plains of the Indian subcontinent [2]. Exposure to weather and environmental conditions varies across different altitudinal zones [3,4]. In addition to the typical temperature lapse rate, substantial variations in wind speed and direction, geographic patterns, humidity, and precipitation differentially influence biodiversity at varying elevations [5].

For both plants and animals, including insects, species richness, and abundance often exhibit a continually declining trend or a mid-elevation peak in response to elevation; however, these patterns vary among taxa and geographical regions [6–11]. The abundance and richness of species are predominantly shaped by abiotic factors associated with elevation [6,12] which impose diversity-limiting constraints at higher altitudes. On an average the Himalaya exhibits a standard temperature lapse rate of 5.65°C per 1000 m [13]. As a result, the higher altitudes significantly alter the duration of the growth period [14] and metabolic rates [15] in insects. Unsuitable environmental conditions hinder insect survival and restrict their altitudinal distribution [16]. Various potential pollinator insect groups are impacted to varying degrees [17], and pollinators communities are involved in the reproduction of mountain angiosperms are likely to vary along the elevation gradients [18].

The composition of pollinator assemblage in insect-pollinated plants exhibits significant temporal and spatial variations among plant communities [19,20] and, in some cases, within a single plant species [21]. For instance, orchids are pollinated by diverse pollinators across different habitats and regions due to variations in pollinator preferences including nectar, pollen, trichomes, fragrance, lipids and resins [22]. These pollinators include morphologically distinct and specialized species of insects, birds, and mammals each adapted to interact with specific flower types for efficient pollen transfer [23,24]. However, pollination is a vital regulating ecosystem service that supports food production and gene flow and ecosystem restoration [25,26]. Despite the expected far-reaching effects of the abiotic environmental factors on plant-pollinator interactions, limited knowledge exists on how these interactions function across altitudinal scales in the Himalayan region. However, some significant research has been conducted that has more precisely addressed the knowledge gap about the spatial variability of pollinators with altitudinal gradient [20,27].

Insects are the primary pollinators of flowering plants in higher altitudes [28]. These insects belong to several orders with Hymenoptera, Lepidoptera, Diptera, Hemiptera and Coleoptera being the most common [29]. Numerous studies worldwide have explored the insect pollinator diversity and pollination ecology of various medicinally important plants [30]. However, few studies have comprehensively assessed how altitude influences the pollinator behaviour and range limits, including

visiting rates, abundance, and reproductive output. Altitudinal gradients provide a valuable opportunity to study pollination ecology at range limits, as declines in pollination and resources availability may limit reproductive success at high altitudes [31,32].

The family Berberidaceae is a diverse group of angiosperms consisting of 19 genera and 700 species worldwide [33], predominantly distributed in the northern hemisphere [34]. *Berberis lycium* Royle, first described by John Forbes Royle in 1837 [35], is a highly valuable shrub and most common member of the Berberidaceae., It is native to the Himalayan region but also thrives in subtropical and temperate climates worldwide [36]. This species has been used in traditional medicine since time immemorial [37] and possesses numerous medicinal properties, including wound healing, hepato-protective, antioxidant, antibacterial, and antihyperlipidemic [38]. In addition, it is traditionally used to treat ophthalmic, liver, skin, cough, and stomach problems. Because of its enormous nutritional and medicinal value, *B. lycium* is subject to overexploitation for consumption and trade, posing a serious threat to its survival in the natural habitat [34]. Furthermore, its habitat is being degraded by anthropogenic activities such as road and infrastructure development, deforestation, and agricultural expansion in hilly areas, leading to a decline in its population [34,39]. According to the IUCN criteria of 2000, *B. lycium* has already been considered as an endangered species in the northwestern Himalaya [40]. Therefore, conservation measures are urgently needed to protect this species from further threats [39]. Flower of *B. lycium* has all the essential traits including fragrance, bright coloration, pollen, and nectar necessitating insect intervention for effective pollen deposition on stigmatic surface [40].

In the northwestern Himalaya, *B. lycium* grows along an altitudinal gradient from 800 to 2200 m asl. Previous studies [41] have suggested that climatic conditions influence floral phenology, pollinatior activity, and life cycle dynamics in *Berberis* species. Given the widespread habitat distribution of *B. lycium* and its reproductive dependence on insect pollinators [30,40]. The present study aims to: (1) document the diversity and variations of prominent insect pollinators with *B. lycium* along altitudinal gradient; (2) compare the pollinator activity across altitudinal gradient; (3) evaluate how pollinator diversity and relative abundance influence the reproductive output of *B. lycium* across altitudinal gradient. We discussed potential hypotheses explaining the observed pollination patterns and their implications in the conservation of this ecologically and medicinally significant plant species.

## 2. Materials and methods

### 2.1. Study area and flowering phenology

The present study was conducted across five different altitudinal sites in the Pir-Panjal mountain range in Jammu and Kashmir of north-western Indian Himalaya. The selected sites included Muradpur (800 m), BGSBU (1150 m), Lower Darhal (1500 m), Upper Darhal B (1850 m), and Dera ki gali (2200 m) and studied from January 2022 to August 2023. The Pir-Panjal mountain range separates the Kashmir valley from the plains of the Jammu division (**Fig 1**). It is the largest mountain range in the lesser Himalaya, with varying average altitudes ranging from 1400 to 4100 m asl [2]. This altitudinal variation provides diverse landscape of mountains and hillocks that surround the Kashmir valley, supporting rich biodiversity, particularly above the 1700 m asl. The study aims to capture this diversity by covering a wide range of habitats, which is reflected in the distinct patterns of biodiversity distribution across altitudes. The flowering and fruiting season of *Berberis lycium* extends from January–July. Flowering begins in January and continues until June, while fruiting ripening starts in May, with full maturation occurring by early August at across the different altitudinal sites.

### 2.2. Sampling and activities of insect pollinators

Documentation and monitoring of insect pollinators were conducted at the study sites between 08:00–18:00 hours from January to June during two consecutive flowering seasons (2022–2023), when *Berberis lycium* was in full bloom. Pollination efficiency was assessed by categorizing insects visits, analyzing their foraging behavior, and recording their flower-visiting

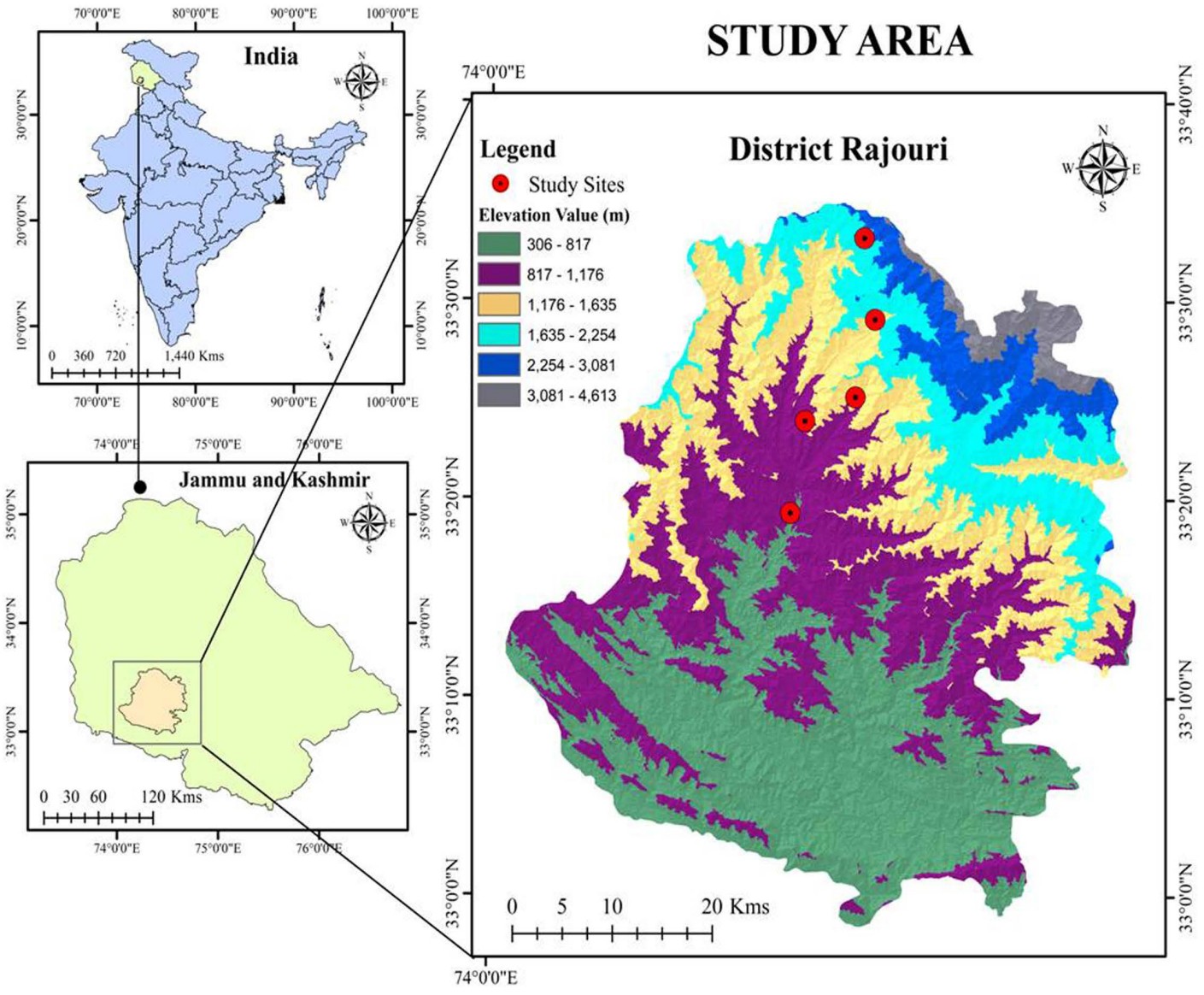

**Fig 1. Map of the study area showing study sites along an altitudinal range of District Rajouri of Jammu and Kashmir, India.**

duration. To determine the role of insects in pollination, notable flower visitors were captured using an insect net and examined under a stereozoom binocular microscope to assess pollen loads on different parts of their bodies. Observations and insect capture were conducted on 30 plants of *B. lycium* per site, with three hours allocated at various times of the day for monitoring. Specimens of insect pollinators were identified at Insect Taxonomy Laboratory (ITL), Department of Zoology, Baba Ghulam Shah Badshah University, Rajouri. On each plant, 15–30 inflorescences were monitored, and the total number of insect visitors observed on 10 plants per site were recorded. During the peak flowering time (mid-February to early May), insect visits were documented at regular one-hour intervals over a 24-hour cycle. To observe nocturnal foragers, a battery-operated torch was used [42]. However, due to absence of nocturnal pollinators on *B. lycium* flowers, observations were ultimately limited to the period between 08:00 and 18:00 hours, with a total observation time of 545 hours).

The contribution of each pollinator to pollen transport was assessed using various pollination indices as below:

**2.2.1. Foraging behaviour (FB).** It was calculated as the time spent by a specific pollinator per visit per inflorescence, counted using a stopwatch [42].

**2.2.2. Foraging speed (FS).** It was measured by counting the number of flowers visited per minute by a pollinator [43] and calculated using the formula:

$$FS = (Fi/di) \times 60$$

where $di$ is the time(in second) recorded by the stopwatch, and $Fi$ is the number of flowers visited during $di$.

**2.2.3. Index of visitation rate (IVR).** IVR is a relative measure of visitor rate, considering both activity rate and frequency of visits. It was calculated using following equation [42]:

$$IVR = F \times AR$$

Where, $F$ represents the proportion of visiting-insect individuals to the total number of insects in the census, whereas $AR$ stands for activity rate, or the average number of flowers visited by a visiting-insect category in a minute.

**2.2.4. Insect visiting efficiency (IVE).** IVE was calculated following [42],

$$IVE = \frac{\text{Number of total flowers visited by insects in one visit}}{\text{Number of total flowers available}}$$

**2.2.5. Potential insect pollinator.** To determine potential pollinators of *B. lycium*, the behaviour of insects was examined at two distinct behavioural decision levels [30].

**(i) Relative abundance** The number of insect pollinators on *B. lycium* was observed on 15–30 inflorescences of 30 plants in each site. **(ii) Sought floral resource:** Insects were grouped based on their primary purpose for visiting *B. lycium* like Collecting Pollen (CP), and Feeding on Pollen (FP), and Feeding on Nectar (FN).

## 2.3. Reproductive output

The reproductive output was assessed by recording the percentage of fruit and seed set for each plant. The percentage fruit set was calculated by determining the average number of flowers, and fruits per plant [40]. It was computed using the following formula:

$$\text{Percentage fruit set} = \left( \frac{\text{Average number of fruit formed per plant}}{\text{Average number of flowers produced per plant}} \right) \times 100$$

The percentage of seed set was determined by calculating the average number of ovules per ovary and the average number of seeds per fruit was calculated [40]. It was calculated using the formula:

$$\text{Percentage seed set} = \left( \frac{\text{Average number of seeds per fruit}}{\text{Average number of ovules per overy}} \right) \times 100$$

## 2.4. Data analysis

The statistical analyses were performed using R software version 4.0.2 (R Core Team 2020; ver. 4.0.2) [44]. Before conducting data analysis, we assessed whether the data for all studied pollination indices (foraging speed, foraging behavior, index of visitation rate, insect visiting efficiency, and relative abundance) met the assumptions of homogeneity of variance and normality of distribution. This was evaluated using Levene's test and the Shapiro-Wilk normality test (implemented via

the 'car' package, ver. 3.1–2)[45]. Since the data did not satisfy the assumptions of homogeneity of variance, and normality, we applied simple linear regression (using the 'Trendline package, ver. 2.0.3) to analyse the relation between insect activity and relative abundance as a function of altitude, with data pooled by insect order. Additionally, linear regression was used to examine the relationship between insect activity and pollen density. Before analysing variations in insect pollination indices along the altitudinal gradient, we reassessed the homogeneity and normality assumptions of the data using Levene's test and the Shapiro-Wilk test. To determine whether there were significant differences in pollination indices across altitudinal gradients, we used an unpaired two-sample t-test (using the 'stats' package, ver. 4.4.0) as the necessary assumptions were met ($p > 0.05$).

## 3. Results

### 3.1. Plant species, floral arrangement and pollination

*Berberis lycium* is an evergreen, spiny, large shrub that grows at altitude ranging from 800 to 2200 m asl. The leaves are coriaceous, slightly obovate-oblong or lanceolate, with a few large, spiny teeth arranged alternately along the stem. The inflorescence consists of corymbose racemes containing 18–24 bright yellow, bracteate, hypogynous, actinomorphic, and hermaphrodite flowers that grow in axillary clusters. Over the calyx, two sporophylls are present. The flower has six light yellow-colored sepals in two whorls, with the outer three being smaller than the inner three. The corolla consists of six bright yellow petals eachwith two nectarines at its base. The androecium is composed of six antipetalous, adnate, and bithecous stamens. The gynoecium consists of a single pistil, divided into the style, stigma, and ovary. The stigma has a central depression and is spherical. While the style is short and empty. Pollination in *B. lycium*is carried out by insects, as the plant possesses key entomophilous traits, including brighter coloured flowers, scent, nectar, and pollen. Insect interaction is essential for the pollen deposition on the stigmatic surface. The fruit is oval-shaped berry that turns bright red or purple when fully ripened. Each fruit contains 2–5 seeds, which range in colour from yellow to pink.

### 3.2. Species composition and activities of insect pollinators

A total of 1,934 insect individuals belonging to five orders, 17 families, and 39 species were recorded visiting *B. lycium* during the flowering period across the altitudinal gradient. At site-1 (800 m), Diptera was the most dominant order with 8 species, followed by the Hymenoptera with four species, Lepidoptera, and Hemiptera with 2 species each, and Coleoptera with one species. At site-2 (1150 m), Diptera was the most dominant order with 11 species, followed by Hymenoptera with 4 species, Hemiptera, Lepidoptera, and Coleoptera with 2 species each. At site-3 (1500 m), Diptera again dominated with 11 species, followed by Hymenoptera and Lepidoptera with 3 species each, and Hemiptera and Coleoptera with one species each. However, at site-4 (1850 m), Lepidoptera was the most dominant order with 8 species, followed by Diptera with 7 species, Hymenoptera with 5 species and, Coleoptera with 2 species. At site-5 (2200 m), Lepidoptera and Hymenoptera both had 7 species, followed by Hymenoptera with 3 species and Coleoptera with 2 species (Table 1–3).

### 3.3. Insect activities

The results of the linear regression showed a statistically non-significant increase ($p > 0.05$) in terms of foraging behaviour of all the studied insect pollinator orders with increasing altitude. The highest increasing rate was observed in Diptera and Hymenoptera (Fig 4A).

A significant decreasing trend ($p < 0.0001$) in the foraging speed was observed only for Lepidoptera (y = 36.4-0.0111x, $R^2 = 0.05$). However, a non-significant increase ($p > 0.05$) was recorded for Hymenoptera and Hemiptera, while a non-significant decrease ($p > 0.05$) was observed for Coleoptera and Diptera with an increasing altitude (Fig 4B).

With increasing altitude, the insect visitation rate showed a significant decrease ($p < 0.05$) in for the Diptera (y = 38.9-0.00677x, $R^2 = 0.03$), whereas Lepidoptera exhibited a significant increase ($p < 0.05$) in insect visitation rate

**Table 1. Showing insect pollinator species belonging to different orders, their foraging activity and altitudinal range (FN = Feeding on Nectar, CP = Collecting Pollen, and FP = Feeding on Pollen).**

| Species | Order | Family | Foraging activity | Altitude (m) |
|---|---|---|---|---|
| *Allograpta* sp. | Hymenoptera | Apidae | FN | 800 |
| *Altica* sp. | Coleoptera | Chrysomelidae | FN, FP | 1150-1500 |
| *Aphidoidea* | Hemiptera | Aphididae | FP | 800-1500 |
| *Apis cerena* | Hymenoptera | Apidae | FN, CP, FP | 1500-1850 |
| *Apis* sp. | Hymenoptera | Apidae | FN, CP, FP | 800-2200 |
| *Bombus haemorrhoidalis* | Hymenoptera | Apidae | FN, CP | 1850 |
| *Bombus* sp. | Hymenoptera | Apidae | FN, CP | 1850-2200 |
| *Bombus trifasciatus* | Hymenoptera | Apidae | FN, CP | 800-1150 |
| *Calliphora vomitoria* | Diptera | Calliphoridae | FN, CP, FP | 800-1150 |
| *Calliphora* sp | Diptera | Calliphoridae | FN, CP, FP | 1150 |
| *Camponotuspennyslyvanicus* | Hymenoptera | Formicidae | FN, CP, FN | 1500-2200 |
| *Celastrina* sp. | Lepidoptera | Lycaenidae | FN | 1850 |
| *Celastrinaargiolus* | Lepidoptera | Lycaenidae | FN | 800-2200 |
| *Chrysotoxumbaphyrum* | Diptera | Syrphidae | FN | 800-1150 |
| *Coccinella septempunctata* | Coleoptera | Coccinellidae | CP | 1850-2200 |
| *Coccinellaundecimpunctata* | Coleoptera | Coccinellidae | CP | 800-2200 |
| *Coelioxys* sp. | Hymenoptera | Megachilidae | FN, CP | 800 |
| *Cyrestisthyodamas* | Lepidoptera | Nymphalidae | FN | 1850-2200 |
| *Dodona durga* | Lepidoptera | Riodinidae | FN | 1500-2200 |
| *Dyscedrus* sp. | Hemiptera | Coreidae | FP | 800-1150 |
| *Episyrphusbalteatus* | Diptera | Syrphidae | FP | 800-2200 |
| *Episyrphysviridaureus* | Diptera | Syrphidae | FP | 1500-2200 |
| *Eristalinustaeniops* | Diptera | Syrphidae | FP | 1550 |
| *Eristaliscerealis* | Diptera | Syrphidae | FP | 1150-2200 |
| *Eristalistenax* | Diptera | Syrphidae | FP | 800-2200 |
| *Eupeodesluniger* | Diptera | Syrphidae | FP | 800-1500 |
| *Formica fusca* | Hymenoptera | Formicidae | FP | 800-1150 |
| *Heliophorus sena* | Lepidoptera | Lycaenidae | FN | 1850-2200 |
| *Heliophorusmoorei* | Lepidoptera | Lycaenidae | FN | 1850-2200 |
| *Heliophorus* sp. | Lepidoptera | Lycaenidae | FN | 1850 |
| *Lucilia* sp. | Diptera | Calliphoridae | FN, CP, FN | 1500-2200 |
| *Musca domestica* | Diptera | Muscidae | FP | 800-2200 |
| *Pierisbrassicae* | Lepidoptera | Pieridae | FN | 2200 |
| *Plecia* sp. | Diptera | Bibionidae | FP | 800-1150 |
| *Praezygaenacaschmirensis* | Lepidoptera | Zygaenidae | CP, FP | 800-1500 |
| *Rhingia* sp. | Diptera | Syrphidae | FN, FP | 1150-1500 |
| *Sarchophaga* sp. | Diptera | Sarchophagidae | FP | 800-1500 |
| *Syritta* sp. | Diptera | Syrphidae | FP | 1500-2200 |
| *Vanessa indica* | Lepidoptera | Nymphalidae | FN | 1850-2200 |

($y = 1.96 + 0.00214x$, $R^2 = 0.08$). However, Coleoptera, Hemiptera, and Hymenoptera showed a non-significant decrease ($p > 0.05$) in IVR (Fig 4C).

Similarly, insect visiting efficiency significantly increased ($p < 0.05$) for the order Hemiptera ($y = 0.0155 + 3.96 \times 10^{-5}x$, $R^2 = 0.31$) with increasing altitude, whereas, Lepidoptera showed a significant decrease ($p < 0.001$) ($y = 0.0941 - 2.37 \times 10^{-5}x$,

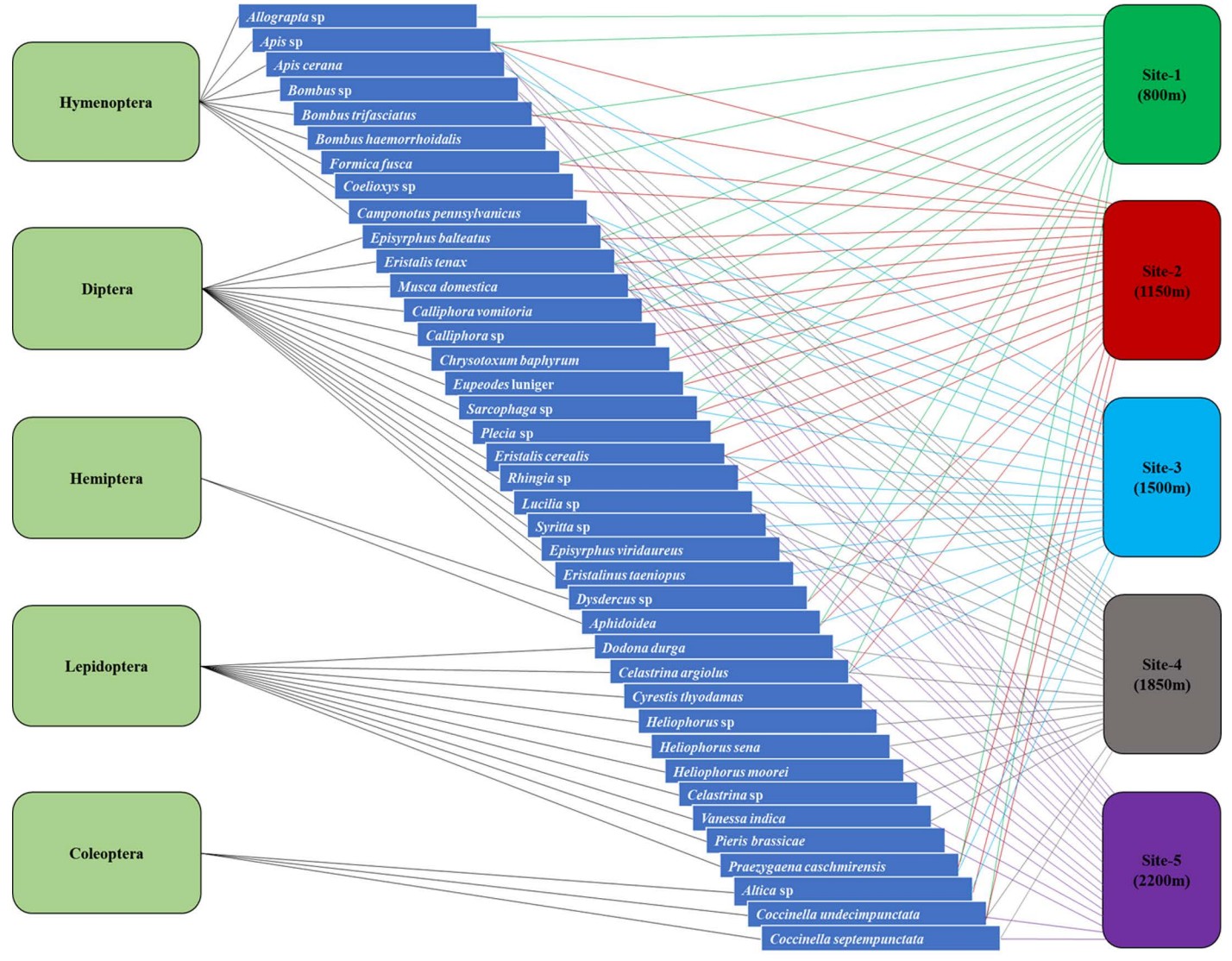

**Fig 2. Interaction of insect pollinators of *Berberis lycium* across study sites along altitudinal gradient.**

$R^2 = 0.21$). However, non-significant decrease ($p > 0.05$) was observed for Coleoptera, while Diptera and Hemiptera exhibited non-significant increase with increasing in altitude (Fig 4D).

A significant decrease ($p < 0.005$) was observed in foraging speed (y = 36.6-0.00566x, $R^2 = 0.03$) and the index of visitation rate (y = 26.4-0.0061x, $R^2 = 0.03$) for overall insect species with increasing attitude. However, foraging behaviour and insect visiting efficiency exhibited a statistically non-significant decrease ($p < 0.05$) (Fig 5).

A statistically significant increase ($p < 0.05$) in relative abundance was observed for Lepidoptera (y = 0.929 + 0.00264x, $R^2 = 0.31$) with increasing altitude. However, Diptera and Hemiptera showed a non-significant decrease ($p < 0.05$), while Coleoptera and Hymenoptera exhibited a non-significant increase (Fig 6).

The average percentage fruit set and seed set decreased with increasing elevation (Fig 7). However, this decrease was non-significant ($p < 0.05$) for both the percentage fruit set (y = 71–0.0154 x, $R^2 = 0.43$) and the percentage seed set (80.9–0.0104 x, $R^2 = 0.42$).

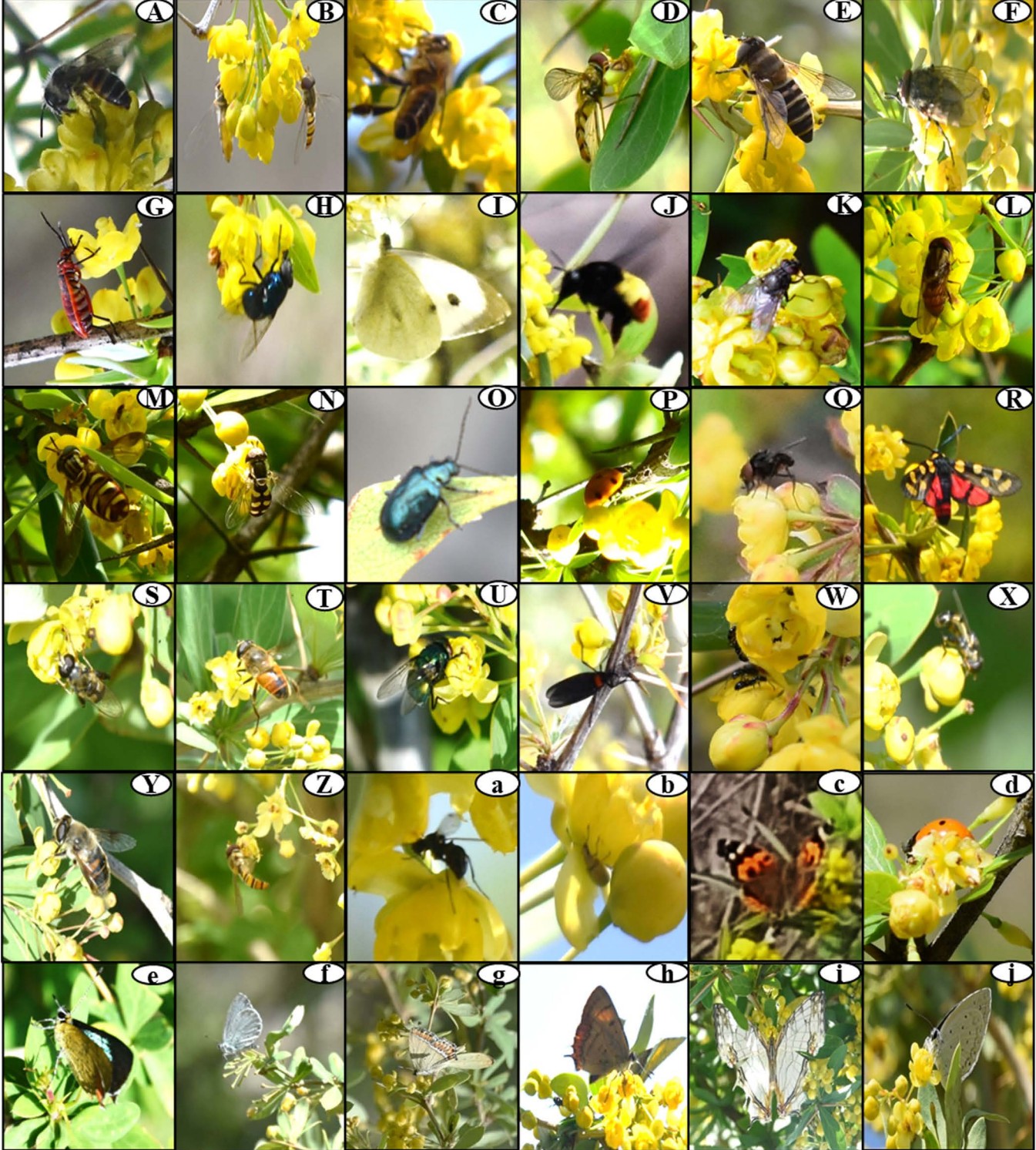

**Fig 3. Representing specimens of various insect pollinators on** *Berberis lycium* **flowers at different altitudinal range of the study area:** A) *Apis* sp., B) *Episyrphus balteatus,* C) *Apis cerena,* D) *Allograpta* sp., E) *Eristalis cerealis,* F) *Musca domestica,* G) *Dysdercus* sp, H) *Calliphora vomitoria,* I) *Pieris brassicae,* J) *Bombus* sp, K) *Calliphora* sp., L) *Rhingia* sp., M) *Chrysotoxum baphyrum* N) *Eupeodes luniger,* O) *Altica* sp., P) *Coccinella undecimpunctata,* Q) *Sarcophaga* sp., R) *Praezygaena caschmirensis,* S) *Syritta* sp., T) *Eristalis tenax,* U) *Lucilia* sp., V) *Plecia* sp., W) *Formica fusca,* X) *Eristalis* sp., Y) *Eristalinus taeniops,* Z) *Episyrphus viridaureus,* a) *Camponotus pennyslyvanicus,* b) *Aphidoidea,* c) *Vanessa*

*indica,* d) *Coccinella septempunctata*, e) *Heliophorus*sp., f) *Celastrina*sp, g) *Heliophorus sena*, h) *Heliophorusmoorei*, i) *Cyrestisthyodamas*, j) *Celastrinaargiolus.*

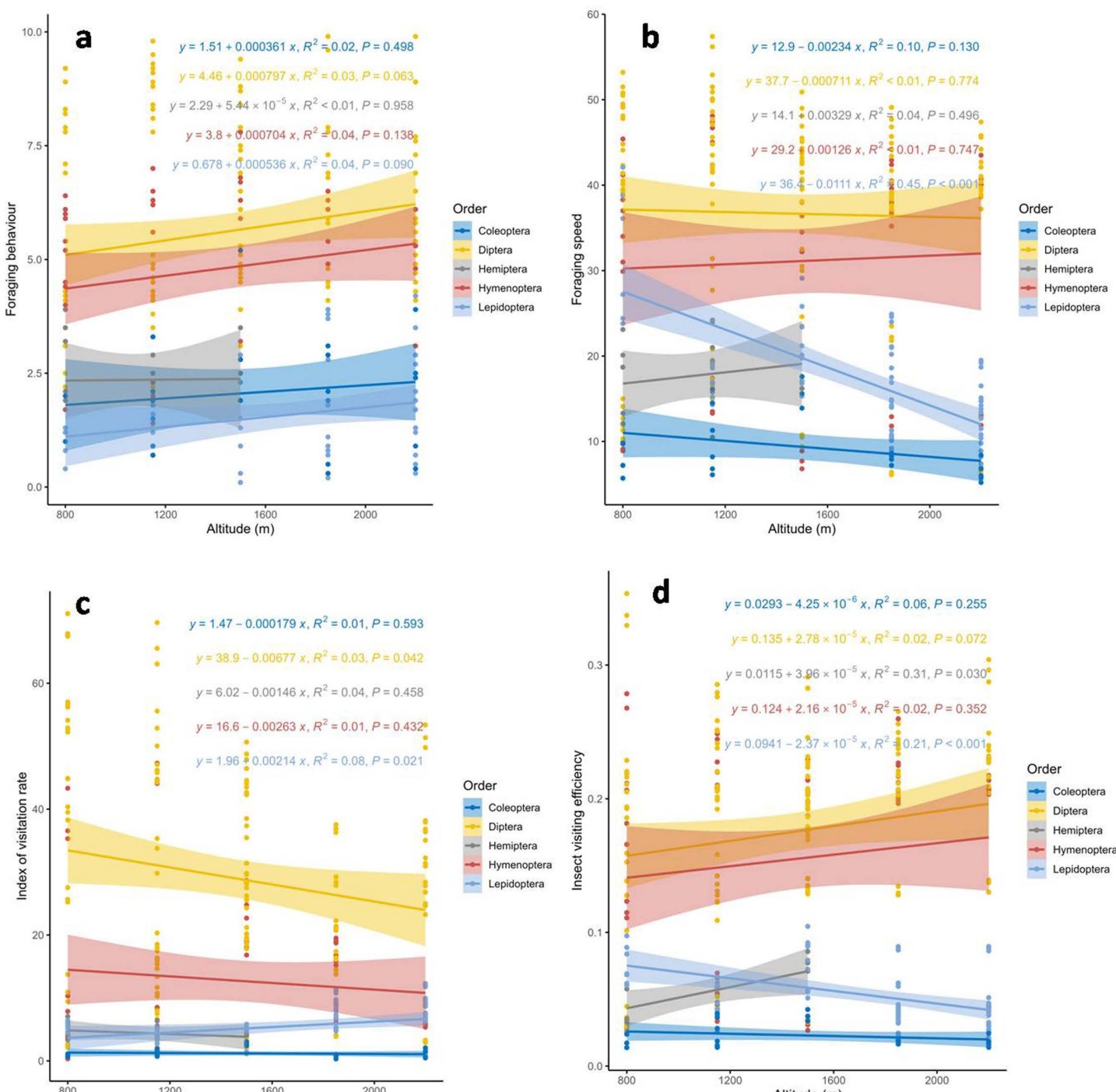

**Fig 4. Activities recorded of different pollinators' orders at different altitudes, A. Foraging behaviour, B. Foraging speed, C. Index of visitation rate, D. Index of visiting efficiency. The best-fitting regression line is represented by the coloured solid line, while the 95% confidence interval of regression line is shown by the shaded areas.**

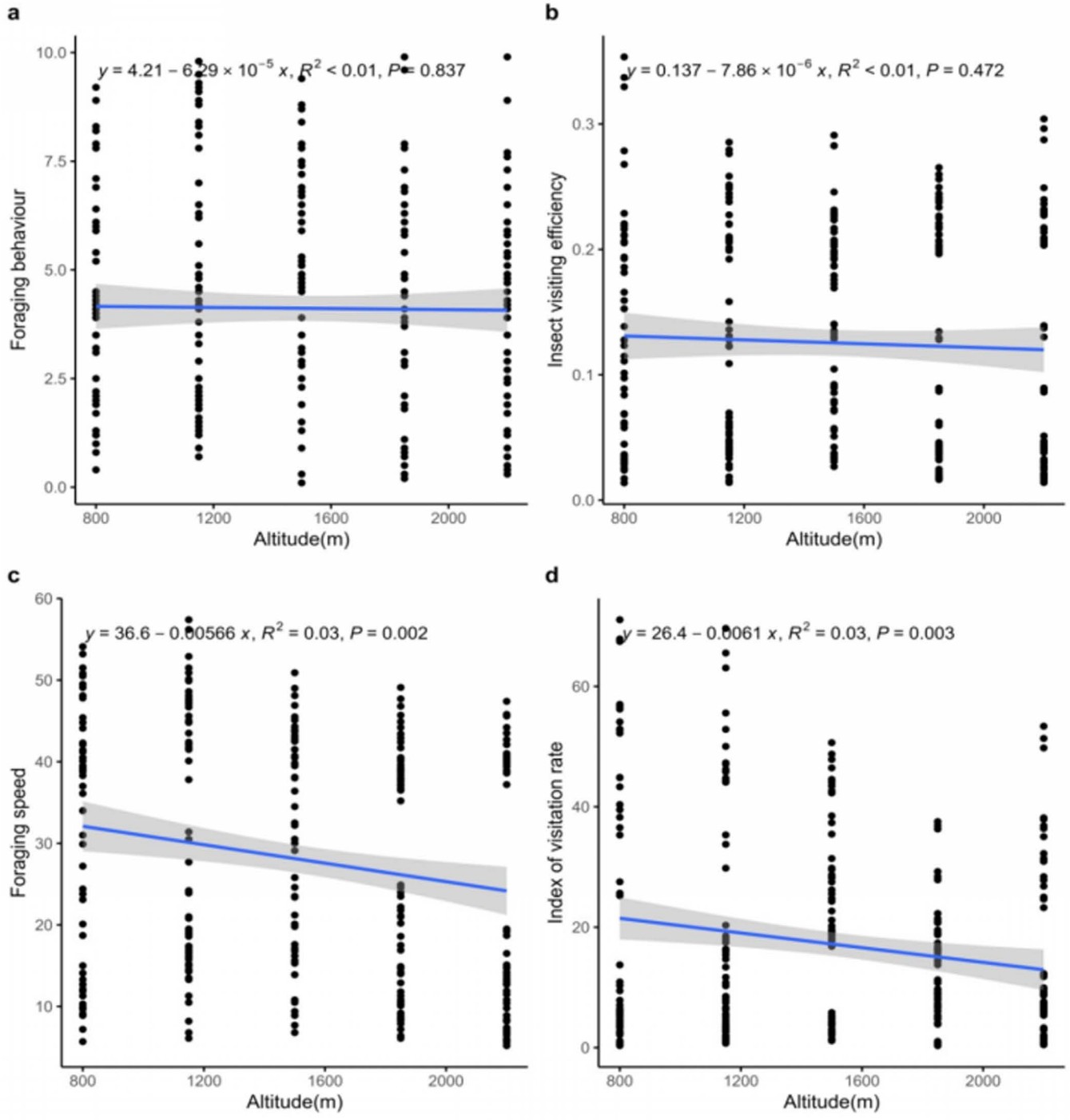

**Fig 5. Activities recorded of combined pollinators' orders at different altitudes, A. Foraging behaviour, B. Foraging speed, C. Index of visitation rate, D. Index of visiting efficiency.** The best-fitting regression line is represented by the blue solid line, while the 95% confidence interval of regression line is shown by the shaded areas.

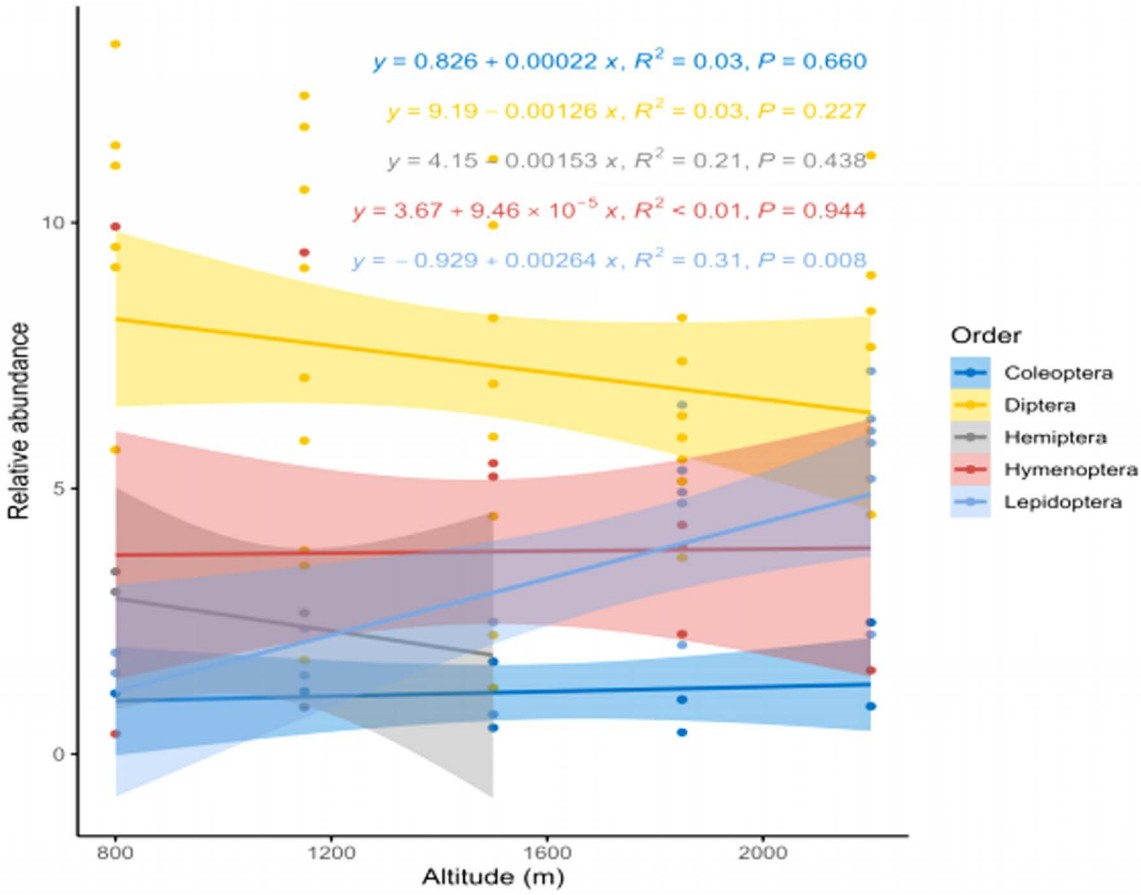

**Fig 6. Relative abundance of different pollinators' orders at different altitudes.** The best-fitting regression line is represented by the coloured solid line, while the 95% confidence interval of regression line is shown by the shaded areas.

## 4. Discussions

In this study, we documented various types of insect pollinators along altitudinal gradients, commonly foraging on flowers of *Berberis lycium*. A total of 1,934 individuals insect pollinators, belonging to 39 species across five orders and 17 families, were observed visiting this plant species during the flowering period across altitudinal gradients. Pooled insect order data from each altitude were used to assess differences in insect activities and abundance across altitudinal zones. Significant variations in the insect pollination indices were observed along five altitudinal gradients. Our results also indicated differences in the relative abundance of insect pollinators and reproductive output along the altitudinal gradients. However, repeated collections on various sampling days at the individual sites provided a broader and more precise assessment of flowering phenology and insect visitors across altitudinal gradients.

This study is the first its kind to document potential insect pollinators of *B. lycium* along an altitudinal gradient in the Pir-Panjal region of Indian western Himalaya. We found that *B. lycium* has a more diverse assemblage of pollinators than previously reported [30,40], likely due to the wide altitudinal range it occupies. In the case of foraging behavior (FB), *Apis* species were the most common foragers across all studied altitudes (Table 1). However, *Musca domestica, Calliphora vomitoria, Sarchophaga* sp. and *Lucilia* sp. exhibited the highest foraging activity among Hymenoptera. The foraging behaviour of all studied insect pollinator orders showed a non-significant increase, with the most pronounced increase

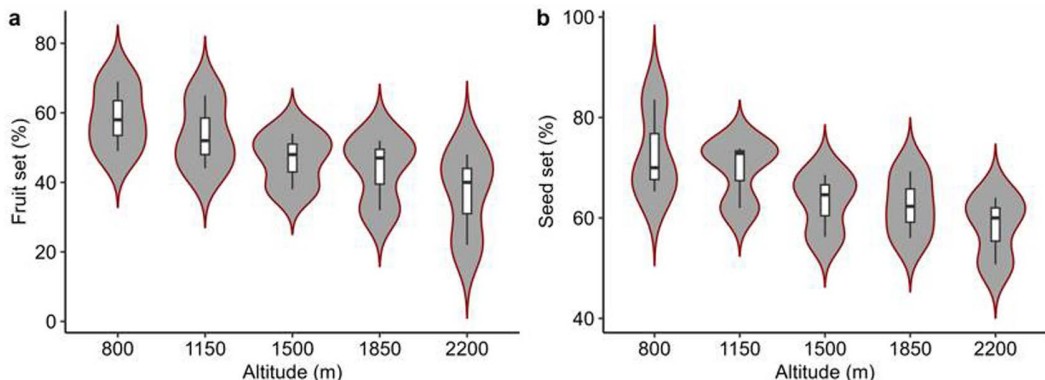

**Fig 7. Change in reproductive output along altitudinal gradient. a) fruit set (PFS; %), b) seed set (PSS; %), The shape in violin plots at 95% confidence, indicates the data of reproductive output.** The violin plots are escorted by whisker boxplots, in which the average reproductive output for each parameter is shown by black lines.

observed in Diptera and Hymenoptera as altitude increased (Fig 4A). Regarding foraging speed (FS), *Calliphora vomitoria, Bombus trifasciatus, Sarchophaga* sp., and *Eupeodes luniger* demonstrated the highest foraging speeds. However, among all the studied orders, a statistically significant decrease in foraging speed was observed only in Lepidoptera (Fig 4B). In the case of insect visiting efficiency (IVE), *Bombus trifasciatus, Calliphora vomitoria,* and *Episyrphus balteatus* exhibited the highest activity levels. When polled data were t analysed, only Diptera showed a significant decrease while Lepidoptera showed a significant increase in insect visiting efficiency (IVE) with increasing altitude (Fig 4C).

For the index of visitation rate (IVR), *Calliphora vomitoria, Eupeodes luniger, Episyrphus balteatus,* and *Eristalis tenax* exhibited the highest activity levels. When data were polled together, only Lepidoptera showed a significant decrease in IVR with increasing altitude (Fig 4D). When considering all four parameters (FB, FS, IVE and, IVR) together, the overall activity rate decreased with increasing altitude. However, only foraging speed (FS) and index of visitation rate (IVR) showed significant results (Fig 5). Additionally, in the case of relative abundance, Lepidoptera showed a statistically significant increase with increase altitude (Fig 6). Our final significant observation was decline in fruit and seed set along the altitudinal gradient (Fig 7).

Observations on major pollinator groups (Hymenoptera, Diptera, Coleoptera, Lepidoptera) may serve as efficient indicators of micro-climatic changes that directly affect their foraging behavioursand nesting places [46]. Most Pollinators belonging to Lepidoptera showed a distinct response to altitude, predominantly occurring within the 1500-2200m range, with few exceptions, signifying their importance in climate change studies [47]. Members of Hemiptera were confined to lower altitudes and were therefor not considered potential pollinators of *B. lycium*. Hymenoptera and Diptera were observed across abroad altitudinal range. At the lowest altitude (800 m), flies (Diptera) were predominant visitors to *B. lycium* (Table 1). Above this altitude, dominant flower-visiting insect orders varied across sites, with no consistent altitudinal pattern except for Lepidoptera. Flies (Diptera) emerged as the most diverse and abundant visitors, playing a crucial role in pollination services for *Berberis lycium* [30,40,41,48]. A decline in their abundance and activity along altitudinal gradient may be linked to reduce reproductive output in this important medicinal plant.

Altitudinal variations in insect taxonomic diversity across altitude have been widely observed on spatial scales [49]. On a finer scale, the number of insect pollinators associated with a specific plant species generally decreases with increasing altitude [8,10,50,51]. For instance, a study on *Betula pubescens* in Norway reported a decline in insect fauna including Heteroptera, Coleoptera, and Homoptera species from 0 to 900 m [50]. Similarly, a study on *Campanula rotundifolia* found lower pollinator diversity and visitation rates at higher altitudes, as highly efficient bumblebees

replaced less efficient solitary bees [52]. These findings align with our results, as all observed insect visitor activities showed a decrease with increasing altitude. However, only foraging speed and the index of visitation rate showed statistically significant results (Fig 5). Our results also indicated that the foraging behaviour of visiting insects increased with altitude across all studied orders (Fig 4A). This variation in foraging behaviour may reflect host plant abundance and richness [16] and could be attributed to a decline in interacting partners at high altitudes [53]. Butterflies and moths are among the most studied insect groups and are highly sensitive to climatic variations, making them ideal for climate change research [47]. This aligns with our findings, as Lepidoptera was the only order to show a significant decrease in foraging speed with increasing altitude (Fig 4B). Our results (supplementary table S1) consistent with the previous findings [54], which reported a positive correlation between Bumblebees diversity and altitude, as well as a negative correlation between altitude and the diversity and visitation rate of non-bumblebees. In the case of insect visiting efficiency, only Lepidoptera and Hemiptera showed significant decreases and increases, respectively (Fig 4D). In temperate regions, anthophilous flies are often the sole pollinators at higher altitudes [55–57] while at lower altitudes, Hymenoptera become more prevalent. In our study, Hemiptera played only a minor role and showed a decreasing tend with increasing altitude.

A great diversity of insects specializes in feeding on floral structures reproductive parts of host plants [42,58,59]. However, many plant species experience a decline in reproductive output with increasing altitudes [8,32,60,61]. The fitness of plant-pollinator interactions is crucial for plant reproductive assurance, floral traits evolution, and breeding systems stability [42]. Breeding systems vary among plant lineages, encompassing complete outcrossers, intermediate outcrossers, and selfers [62,63]. Lineages undergoing repeated selfing often suffer from inbreeding depression, which can lead to fitness loss. Therefore, insect pollination plays a key role in ensuring both the nutritional and physical quality of plant reproductive structures [64]. Although biotic interactions significantly influence plant range expansion, pollination syndrome is an essential evolutionary mechanism maintaining ecological boundaries [65]. The decline in insect pollinators is likely to affect plant reproductive traits of plants growing in vulnerable habitats. For instance, early flowering, and self-incompatible plants rely more on pollinators than mid-flowering, self-compatible species, making them particularly susceptible to pollinator declines. The high dependence of *B. lycium* on pollinators, as observed in our study suggests that if the plant cannot adjust its reproductive strategies to compensate for pollinators declines, its seed production will be severely impacted by climate change [66].

## 5. Conclusion

Our findings provide valuable insight for conservation biologists studying plant-pollinator interactions in the context of global climate change, which is predicted to alter the spatio-temporal distribution of anthophilous insects. Butterflies and moths (Lepidoptera) emerged as the most sensitive pollinators, showing significant variations in activity along the altitudinal gradient. We documented the diversity of anthophilous insects associated with *B. lycium* pollination across different altitudes. Our study, in line with global trends, observed a decline in the overall abundance and activity of *B. lycium* pollinators with increasing in altitude, which correlated negatively with its reproductive output. Given *B. lycium*'s high reliance on pollinators, if the plant fails to adjust its reproductive strategies in response to pollinator declines, its seed production will be significantly impacted by climate change.

Based on our findings, the study suggests the conserve key pollinators, with a particular focus on Lepidoptera, by prioritizing the protection of butterfly and moth habitats at higher altitudes, ii) restore and connect habitats by maintaining plant diversity along altitudinal gradients to provide continuous foraging resources, ensuring stable plant-pollinator interactions, iii) promote climate-resilient pollination strategies that support diverse pollinator communities and enhance ecosystem resilience to environmental changes, iv) establish long-term monitoring programmes to study pollination dynamics, v) engage local communities in pollinator conservation and, vi) integrate pollinator conservation into climate adaptation strategies to ensure the development of pollinator-friendly environments.

## Supporting information

**Supplementary table S1. Showing Foraging Behaviour, Foraging Speed, Insect Visiting Efficiency, Index of Visitation Rate, Density and Relative abundance of various insects' individual species and orders at different altitudes.** (DOCX)

The authors are thankful to Dr. Sajad H. Parey, Department of Zoology, BGSB University, Rajouri for insect identification. Besides, first author (Ms. Nahila Anjum) is also thankful to CSIR-HRDG, New Delhi for providing Junior and Senior Research Fellowship (File no. 09/1172(0005)/2019-EMR-I). We also like to thank the anonymous reviewers for critically reviewing the manuscript.

## Author contributions

**Conceptualization:** Nahila Anjum, Susheel Verm, Mohd Hanief.

**Data curation:** Nahila Anjum.

**Formal analysis:** Nahila Anjum, Sajid Khan.

**Funding acquisition:** Nahila Anjum.

**Investigation:** Susheel Verm, Kailash S. Gaira, Balwant Rawat, Nakul Chettri, Mohd Hanief.

**Methodology:** Nahila Anjum, Sajid Khan.

**Project administration:** Susheel Verm, Nakul Chettri, Mohd Hanief.

**Resources:** Nahila Anjum, Susheel Verm, Mohd Hanief.

**Software:** Nahila Anjum, Sajid Khan.

**Supervision:** Nakul Chettri, Mohd Hanief.

**Validation:** Susheel Verm, Kailash S. Gaira, Balwant Rawat, Nakul Chettri, Mohd Hanief.

**Visualization:** Balwant Rawat.

**Writing – original draft:** Nahila Anjum, Sajid Khan.

**Writing – review & editing:** Nahila Anjum, Sajid Khan, Susheel Verm, Kailash S. Gaira, Balwant Rawat, Nakul Chettri, Mohd Hanief.

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
