## [Decision Letter · Decision Letter 0]

10 Oct 2024

PONE-D-24-38069Plant-Pollinator Interactions along the Altitudinal Gradient in Berberis lycium Royle: An Endangered Medicinal Plant Species of the Himalayan RegionPLOS ONE

Dear Dr. Chettri,

Thank you for submitting your manuscript to PLOS ONE. After careful consideration, we feel that it has merit but does not fully meet PLOS ONE’s publication criteria as it currently stands. Therefore, we invite you to submit a revised version of the manuscript that addresses the points raised during the review process.

We look forward to receiving your revised manuscript.

Kind regards,

Taslima Sheikh

Academic Editor

PLOS ONE

Journal requirements: When submitting your revision, we need you to address these additional requirements.

2. Please match your authorship list in your manuscript file and in the system. 3. In your Methods section, please provide additional information regarding the permits you obtained for the work. Please ensure you have included the full name of the authority that approved the field site access and, if no permits were required, a brief statement explaining why. 4. We note that you have indicated that there are restrictions to data sharing for this study. For studies involving human research participant data or other sensitive data, we encourage authors to share de-identified or anonymized data. However, when data cannot be publicly shared for ethical reasons, we allow authors to make their data sets available upon request. For information on unacceptable data access restrictions, please see http://journals.plos.org/plosone/s/data-availability#loc-unacceptable-data-access-restrictions.  Before we proceed with your manuscript, please address the following prompts:

b) If there are no restrictions, please upload the minimal anonymized data set necessary to replicate your study findings to a stable, public repository and provide us with the relevant URLs, DOIs, or accession numbers. Please see http://www.bmj.com/content/340/bmj.c181.long for guidelines on how to de-identify and prepare clinical data for publication. For a list of recommended repositories, please see https://journals.plos.org/plosone/s/recommended-repositories. You also have the option of uploading the data as Supporting Information files, but we would recommend depositing data directly to a data repository if possible. Please update your Data Availability statement in the submission form accordingly.

5. In the online submission form, you indicated that [The data underlying the results presented in the study are available from the first author on request]. 

All PLOS journals now require all data underlying the findings described in their manuscript to be freely available to other researchers, either 1. In a public repository, 2. Within the manuscript itself, or 3. Uploaded as supplementary information.This policy applies to all data except where public deposition would breach compliance with the protocol approved by your research ethics board. If your data cannot be made publicly available for ethical or legal reasons (e.g., public availability would compromise patient privacy), please explain your reasons on resubmission and your exemption request will be escalated for approval. 

Additional Editor Comments:

Few species of butterflies are not rightly identified, kindly use Parey & Sheikh, 2021 book for the identification of butterflies in the region...

Authors are requested to follow up the comments of both reviewers and resubmit...

Reviewers' comments:

Reviewer's Responses to Questions

**Comments to the Author**

1. Is the manuscript technically sound, and do the data support the conclusions?

Reviewer #1: Partly

Reviewer #2: Yes

2. Has the statistical analysis been performed appropriately and rigorously? 

Reviewer #1: Yes

Reviewer #2: Yes

3. Have the authors made all data underlying the findings in their manuscript fully available?

Reviewer #1: Yes

Reviewer #2: Yes

4. Is the manuscript presented in an intelligible fashion and written in standard English?

Reviewer #1: Yes

Reviewer #2: Yes

5. Review Comments to the Author

Reviewer #1: Dear Authors,

This paper is technically in poor form. The identification of insect pollinators and their pictures must be perfect enough as this is the base of all your further statistical methodology. The R-package has many functions that could have been added further to support your hypothesis.

What I believe and with my field experiences I suspect how so many pollinators in extreme cold conditions from January to June could be observed at an altitudinal range of 2200 m amsl as Pirpanjal range usually remains fully accumulated with snow upto June.

More works needs to be done for this paper for being considered for its publication in PLOS ONE.

Reviewer #2: The manuscript sound good with the adequate information. however, few minor corrections to be added and if possible make a separate table to indicate the peak pollination timing of different order of insects which will provide additional information to the readers and great insights for the readers to take up similar studies of plant-pollinator interactions. The scientific names listed in the table were clubbed with genus and species, make sure to separate them and include the authority names beside each species.

6. PLOS authors have the option to publish the peer review history of their article (what does this mean? ). If published, this will include your full peer review and any attached files.

**Do you want your identity to be public for this peer review?** For information about this choice, including consent withdrawal, please see our Privacy Policy .

Reviewer #1: No

Reviewer #2: **Yes: ** Dr. Muddasar

---

## [Author Response · Author response to Decision Letter 1]

5 Feb 2025

Compliance to editors’ comments

Reviewer#1

Comment 1: The terrestrial ecosystem especially Mountains.

Response: The sentence has beencorrected as suggested. (Page no. 2, line no.22)

Comment 2: and.

Response: Corrected. (Page no. 2, line no. 29)

Comment 3: Mountains alone are not considered as ecosystem, it’s a broad term, and composed of terrestrial ecosystems among which Mountains are one of the components.

Response: Corrected. (Page no. 2, line no. 44)

Comment 4: remove.

Response: Extra full stop has been removed from sentence. (Page no. 3, line no. 77)

Comment 5: done at.

Response: Corrected. (Page no. 4, line no. 121)

Comment 6: visiting.

Response: Corrected. (Page no. 4, line no. 124)

Comment 7: stereo zoom binocular microscope.

Response: Corrected. (Page no. 4, line no. 125)

Comment 8: A total of.

Response: Corrected. (Page no. 6, line no. 200)

Comment 9: Documented on.

Response: Corrected. (Page no. 6, line no. 200)

Comment 10: Arrange the species alphabetically or family-wise.

Response: Thank you. The names of the species have been arranged alphabetically. (Page no. 8, Table no. 1)

Comment 11: Separate genus and species.

Response: Genus has been separated from species. (Page no. 9, Table no. 1).

Reviewer#2

Comment 1: In acknowledgment section it is evident that first author has received funding for this work from CSIR-UGC. Must be mentioned here as well.

Response: Thank you for this comment. The funding has been mentioned in the online submission portal.

Comment 2: As per ICZN rule it is important to mention the year of the publication at least once in the paper.

Response: The year of publication has been mentioned (Zainab et al., 2023). (Page no. 3, line no. 82)

Comment 3: Authors need to review the literature of Insect Pollinator interactions and the environmental preference along the attitudinal range as they have missed many important works that have been carried out on insect pollinators along the attitudinal gradient of Himalaya e.g Rather et al. 2023; Singh et al. 2022; Arya and Badoni, 2023; Shrestha et al. 2013 etc.

Response: Thank you for valuable comment. The literature review has been revised and the suggested literature have been cited accordingly (Page no. 2)

Comment 4: Cite this sentence.

Response: The sentence has been cited as suggested. (Page no. 2, line no 47)

Comment 5: Geographic pattern.

Response: Corrected. (Page no. 2, line no. 49)

Comment 6: In my opinion it is not poorly studied. Good amount of work has been carried out. Authors again need to thoroughly review the published work.

Response: The sentence has been revised with relevant citations. ((Page no. 3, line no. 69-71)

Comment 7: Cite the most recent reference for the global species number of Berberidaceae.

Response: The most recent reference has been cited (Hsieh et al., 2022). (Page no. 3, line no. 80)

Comment 8: Cite it.

Response: The sentence has been cited. (Page no. 3, line no. 81)

Comment 9: Cite.

Response: The sentence has been cited. (Page no. 3, line no. 83)

Comment 10: Cite.

Response: We have added citation. (Page no. 3, line no. 84)

Comment 11: Citation is lacking in most of the concrete statements.

Response: We have revised and added citations. (Page no. 3, line no. 86, 89, 93)

Comment 12: Authors must add note on the pollination mechanism of B. lycium whether the plant is self-pollinated or cross-pollinated. This will be much helpful to the readers.

Response: Brief note on pollination mechanism of B. lycium were already added in the MS as the plant is cross-pollinated (entomophilous). (Page no. 3, line no. 93)

Comment 13: revealed.

Response: Corrected. (Page no. 3, line no. 97)

Comment 14: variation? rewrite this sentence. We revised the sentence.

Response: We have revised the sentence. (Page no. 4, line no. 101)

Comment 15: From January upto May the Pirpanjal is having extreme cold conditions with snowfall all around. Insect activity all the high-altitude regions generally starts from June-September. Authors must provide the name of the localities, along with the altitude to make this study more authentic and relevant.

Response: The authors are thankful for this comment. The name of the localities along with altitude has been added in the text. (Page no. 4, line no. 110)

Comment 16: Did the authors submit the collection in this Museum. If yes they must provide the accession number of each sample.

Response: The collection is not submitted yet in the Museum. (Page no. 5, line no. 128)

Comment 17: The noctural study must be cited? This procedure seems inadequate since the light could have attracted pollinators towards it and thus error in data.

Response: The nocturnal study has been cited. We acknowledge your concern about the attraction of nocturnal insects to light. However, we only observed insect pollinators that were already present on B. lycium flowers, if any. Due to the absence of nocturnal insect visitation on the flowers, we confined our study accordingly. (Page no. 4, line no. 132)

Comment 18: How modularity changes with different altitudes can provide more meanings to this paper.

Response: We agree with your valuable comment that modular changes across different altitudes are important in establishing a meaningful context for the work presented. Therefore, key parameters reflecting this modularity have been highlighted in the first and second paragraphs of the introduction section.

Comment 19: Give details about the R package used for the analysis.

Response: Thank you for this comment. The details of the packages used has been added in the data analysis section in the highlighted text. (Page no. 6, line no. 175, 177, and 183)

Comment 20: I wonder, is this the result of the current study or it is already observed phenological traits of this particular plant species. This part must be omitted from the Results section as its already studied traits of B. lycium.

Response: The phenological traits of B. lycium have already been observed in previous studies. However, in the current study, we not only observed these traits again but also examined how they varied along the altitudinal gradient. Therefore, we have included phenological traits in this study. (Page no. 6, line no. 189)

Comment 21: How they determine this number? Generally, we use the mark and recapture method. How authors determined it.

Response: We determined this number by randomly selecting 15-30 inflorescences on each of 30 B. lycium plants at each site, with the plants spaced approximately 5 meters apart. The plants were visited for about an hour, with intervals of 5 minutes between each plant, once a week on sunny days. A visual count of individual insects was conducted for 2 minutes on each selected plant, and this was repeated during each sampling hour of the day (Badoni and Arya, 2020). (Page no. 7)

Comment 22: Not all samples have been studied upto the species level. Some samples are identified upto genus level only. Add upto the species level identification also.

Response: Thank you for this comment. We made every effort to identify all the captured specimens to the species level. However, some insects could only be identified to the genus level. (Page no. 6)

Comment 23: What is the ecological significance of this study. Since in Fig 2 you have shown the insect network associated with the B. lycium along the altitudinal gradient. In order to analyse and interpret this study authors must take the help from already published work on network analysis like Rather et al. (2023); Basu et al. (2022), and similar published work.

Response: We greatly appreciate the suggested literature (Rather et al. 2023; Basu et al. 2022). However, in this research, we observed how the single plant species (B. lycium) is pollinated in different months at various altitudes, and which insect species are either common or unique across these altitudes by studying their activities. (Page no. 8, Figure no. 2)

Comment 24: They collect pollen and feed on nectar also.

Response: Thank you, we have revised the same. (Page no. 8, Table no. 1).

Comment 25: See the above comment on Bombus.

Response: Thank you, we have revised the same. (Page no. 8, Table no. 1).

Comment 26: We generally co-relate our work with the previous works on the insect population declines. Did authors find any such previous work on the decline of insect pollinators along the study region.

Response: Thank you. We were unable to find any previous work in the studied region that reports a decline in insect populations with increasing latitude (Page no. 17)

Comment 27: In online portal data submission, authors have submitted that no funding was available for this study.

Response: Thank you for this comment. The funding support has been acknowledged, but it is not currently available. Therefore, we have submitted the work without any available funding. (Page no. 18)

Comment 28: Most of the insect pictures are blurred in this paper and it is impossible to identify them upto the species level. Authors must use good quality pictures in this paper as whole of the data analyses are dependent on the proper species identification.

Response: Thank you for this comment. The insect pictures have been replaced with higher-quality images. We made every effort to capture clear pictures of insects while they were foraging. However, the captured insects were identified in the laboratory by expert taxonomists. (Page no. 10, Figure No. 3).

End

---

## [Editor Report · Decision Letter 1]

21 Feb 2025

PONE-D-24-38069R1Plant-Pollinator Interactions along the Altitudinal Gradient in Berberis lycium Royle: An Endangered Medicinal Plant Species of the Himalayan RegionPLOS ONE

Dear Dr. Chettri,

Thank you for submitting your manuscript to PLOS One. After careful consideration, we feel that it has satisfied our scientific requirements for publication.

However, our editorial team have significant concerns about the grammar, usage, and overall readability of the manuscript. PLOS One requires that published manuscripts use language which is 'clear, correct, and unambiguous', see our criteria for publication at https://journals.plos.org/plosone/s/criteria-for-publication#loc-5. We therefore request that you revise the text to fix the grammatical errors and improve the overall readability of the text.

We suggest you have a fluent English-language speaker thoroughly copyedit your manuscript for language usage, spelling, and grammar. If you do not know anyone who can do this, you may wish to consider employing a professional scientific editing service.

Whilst you may use any professional scientific editing service of your choice, PLOS has partnered with both American Journal Experts (AJE) and Editage to provide discounted services to PLOS authors. Both organizations have experience helping authors meet PLOS guidelines and can provide language editing, translation, manuscript formatting, and figure formatting to ensure your manuscript meets our submission guidelines. To take advantage of our partnership with AJE, visit the AJE website (https://www.aje.com/go/plos/) for a 15% discount off AJE services. To take advantage of our partnership with Editage, visit the Editage website (www.editage.com) and enter referral code PLOSEDIT for a 15% discount off Editage services. If the PLOS editorial team finds any language issues in text that either AJE or Editage has edited, the service provider will re-edit the text for free.

Please note that we will not be able to proceed with publication of your manuscript until the concerns above are addressed.

* A copy of your manuscript showing your changes by either highlighting them or using track changes (uploaded as a supporting information file)

* A clean copy of the edited manuscript (uploaded as the new manuscript file)

We look forward to receiving your revised manuscript.

Kind regards,

Miquel Vall-llosera Camps

Senior Staff Editor

PLOS One

on behalf of

Taslima Sheikh

Academic Editor

PLOS One
---

## [Author Response · Author response to Decision Letter 2]

9 Mar 2025

Compliance to editors’ comments

Academic Editors Comments

Thank you for submitting your manuscript to PLOS One. After careful consideration, we feel that it has satisfied our scientific requirements for publication.

Response: Thank you very much for this positive note. Grateful for your consistent support.

However, our editorial team have significant concerns about the grammar, usage, and overall readability of the manuscript. PLOS One requires that published manuscripts use language which is 'clear, correct, and unambiguous', see our criteria for publication at https://journals.plos.org/plosone/s/criteria-for-publication#loc-5. We therefore request that you revise the text to fix the grammatical errors and improve the overall readability of the text.

We suggest you have a fluent English-language speaker thoroughly copyedit your manuscript for language usage, spelling, and grammar. If you do not know anyone who can do this, you may wish to consider employing a professional scientific editing service.

Response: Thank you very much for the suggestion. We have now thoroughly revised the manuscript with inputs from professional language editor. All the changes made are reflected in track change mode. A clean version also included and following are the detail about the response to the reviewers’ comments.

Reviewer#1

Comment 1: The terrestrial ecosystem especially Mountains.

Response: The sentence has beencorrected as suggested. (Page no. 2, line no.22)

Comment 2: and.

Response: Corrected. (Page no. 2, line no. 29)

Comment 3: Mountains alone are not considered as ecosystem, it’s a broad term, and composed of terrestrial ecosystems among which Mountains are one of the components.

Response: Corrected. (Page no. 2, line no. 44)

Comment 4: remove.

Response: Extra full stop has been removed from sentence. (Page no. 3, line no. 77)

Comment 5: done at.

Response: Corrected. (Page no. 4, line no. 121)

Comment 6: visiting.

Response: Corrected. (Page no. 4, line no. 124)

Comment 7: stereo zoom binocular microscope.

Response: Corrected. (Page no. 4, line no. 125)

Comment 8: A total of.

Response: Corrected. (Page no. 6, line no. 200)

Comment 9: Documented on.

Response: Corrected. (Page no. 6, line no. 200)

Comment 10: Arrange the species alphabetically or family-wise.

Response: Thank you. The names of the species have been arranged alphabetically. (Page no. 8, Table no. 1)

Comment 11: Separate genus and species.

Response: Genus has been separated from species. (Page no. 9, Table no. 1).

Reviewer#2

Comment 1: In acknowledgment section it is evident that first author has received funding for this work from CSIR-UGC. Must be mentioned here as well.

Response: Thank you for this comment. The funding has been mentioned in the online submission portal.

Comment 2: As per ICZN rule it is important to mention the year of the publication at least once in the paper.

Response: The year of publication has been mentioned (Zainab et al., 2023). (Page no. 3, line no. 82)

Comment 3: Authors need to review the literature of Insect Pollinator interactions and the environmental preference along the attitudinal range as they have missed many important works that have been carried out on insect pollinators along the attitudinal gradient of Himalaya e.g Rather et al. 2023; Singh et al. 2022; Arya and Badoni, 2023; Shrestha et al. 2013 etc.

Response: Thank you for valuable comment. The literature review has been revised and the suggested literature have been cited accordingly (Page no. 2)

Comment 4: Cite this sentence.

Response: The sentence has been cited as suggested. (Page no. 2, line no 47)

Comment 5: Geographic pattern.

Response: Corrected. (Page no. 2, line no. 49)

Comment 6: In my opinion it is not poorly studied. Good amount of work has been carried out. Authors again need to thoroughly review the published work.

Response: The sentence has been revised with relevant citations. ((Page no. 3, line no. 69-71)

Comment 7: Cite the most recent reference for the global species number of Berberidaceae.

Response: The most recent reference has been cited (Hsieh et al., 2022). (Page no. 3, line no. 80)

Comment 8: Cite it.

Response: The sentence has been cited. (Page no. 3, line no. 81)

Comment 9: Cite.

Response: The sentence has been cited. (Page no. 3, line no. 83)

Comment 10: Cite.

Response: We have added citation. (Page no. 3, line no. 84)

Comment 11: Citation is lacking in most of the concrete statements.

Response: We have revised and added citations. (Page no. 3, line no. 86, 89, 93)

Comment 12: Authors must add note on the pollination mechanism of B. lycium whether the plant is self-pollinated or cross-pollinated. This will be much helpful to the readers.

Response: Brief note on pollination mechanism of B. lycium were already added in the MS as the plant is cross-pollinated (entomophilous). (Page no. 3, line no. 93)

Comment 13: revealed.

Response: Corrected. (Page no. 3, line no. 97)

Comment 14: variation? rewrite this sentence. We revised the sentence.

Response: We have revised the sentence. (Page no. 4, line no. 101)

Comment 15: From January upto May the Pirpanjal is having extreme cold conditions with snowfall all around. Insect activity all the high-altitude regions generally starts from June-September. Authors must provide the name of the localities, along with the altitude to make this study more authentic and relevant.

Response: The authors are thankful for this comment. The name of the localities along with altitude has been added in the text. (Page no. 4, line no. 110)

Comment 16: Did the authors submit the collection in this Museum. If yes they must provide the accession number of each sample.

Response: The collection is not submitted yet in the Museum. (Page no. 5, line no. 128)

Comment 17: The noctural study must be cited? This procedure seems inadequate since the light could have attracted pollinators towards it and thus error in data.

Response: The nocturnal study has been cited. We acknowledge your concern about the attraction of nocturnal insects to light. However, we only observed insect pollinators that were already present on B. lycium flowers, if any. Due to the absence of nocturnal insect visitation on the flowers, we confined our study accordingly. (Page no. 4, line no. 132)

Comment 18: How modularity changes with different altitudes can provide more meanings to this paper.

Response: We agree with your valuable comment that modular changes across different altitudes are important in establishing a meaningful context for the work presented. Therefore, key parameters reflecting this modularity have been highlighted in the first and second paragraphs of the introduction section.

Comment 19: Give details about the R package used for the analysis.

Response: Thank you for this comment. The details of the packages used has been added in the data analysis section in the highlighted text. (Page no. 6, line no. 175, 177, and 183)

Comment 20: I wonder, is this the result of the current study or it is already observed phenological traits of this particular plant species. This part must be omitted from the Results section as its already studied traits of B. lycium.

Response: The phenological traits of B. lycium have already been observed in previous studies. However, in the current study, we not only observed these traits again but also examined how they varied along the altitudinal gradient. Therefore, we have included phenological traits in this study. (Page no. 6, line no. 189)

Comment 21: How they determine this number? Generally, we use the mark and recapture method. How authors determined it.

Response: We determined this number by randomly selecting 15-30 inflorescences on each of 30 B. lycium plants at each site, with the plants spaced approximately 5 meters apart. The plants were visited for about an hour, with intervals of 5 minutes between each plant, once a week on sunny days. A visual count of individual insects was conducted for 2 minutes on each selected plant, and this was repeated during each sampling hour of the day (Badoni and Arya, 2020). (Page no. 7)

Comment 22: Not all samples have been studied upto the species level. Some samples are identified upto genus level only. Add upto the species level identification also.

Response: Thank you for this comment. We made every effort to identify all the captured specimens to the species level. However, some insects could only be identified to the genus level. (Page no. 6)

Comment 23: What is the ecological significance of this study. Since in Fig 2 you have shown the insect network associated with the B. lycium along the altitudinal gradient. In order to analyse and interpret this study authors must take the help from already published work on network analysis like Rather et al. (2023); Basu et al. (2022), and similar published work.

Response: We greatly appreciate the suggested literature (Rather et al. 2023; Basu et al. 2022). However, in this research, we observed how the single plant species (B. lycium) is pollinated in different months at various altitudes, and which insect species are either common or unique across these altitudes by studying their activities. (Page no. 8, Figure no. 2)

Comment 24: They collect pollen and feed on nectar also.

Response: Thank you, we have revised the same. (Page no. 8, Table no. 1).

Comment 25: See the above comment on Bombus.

Response: Thank you, we have revised the same. (Page no. 8, Table no. 1).

Comment 26: We generally co-relate our work with the previous works on the insect population declines. Did authors find any such previous work on the decline of insect pollinators along the study region.

Response: Thank you. We were unable to find any previous work in the studied region that reports a decline in insect populations with increasing latitude (Page no. 17)

Comment 27: In online portal data submission, authors have submitted that no funding was available for this study.

Response: Thank you for this comment. The funding support has been acknowledged, but it is not currently available. Therefore, we have submitted the work without any available funding. (Page no. 18)

Comment 28: Most of the insect pictures are blurred in this paper and it is impossible to identify them upto the species level. Authors must use good quality pictures in this paper as whole of the data analyses are dependent on the proper species identification.

Response: Thank you for this comment. The insect pictures have been replaced with higher-quality images. We made every effort to capture clear pictures of insects while they were foraging. However, the captured insects were identified in the laboratory by expert taxonomists. (Page no. 10, Figure No. 3).

End

---

## [Editor Report · Decision Letter 2]

11 Mar 2025

Plant-Pollinator Interactions Along the Altitudinal Gradient in Berberis lycium Royle: An Endangered Medicinal Plant of the Himalayan Region

PONE-D-24-38069R2

Dear Dr. Chettri,

We’re pleased to inform you that your manuscript has been judged scientifically suitable for publication and will be formally accepted for publication once it meets all outstanding technical requirements.

Kind regards,

Taslima Sheikh

Academic Editor

PLOS ONE
---

## [Editor Report · Acceptance letter]

PONE-D-24-38069R2

PLOS ONE

Dear Dr. Chettri,

I'm pleased to inform you that your manuscript has been deemed suitable for publication in PLOS ONE. Congratulations! Your manuscript is now being handed over to our production team.

Kind regards,

on behalf of

Dr. Taslima Sheikh

Academic Editor

PLOS ONE